# Visualization of translation termination intermediates trapped by the Apidaecin 137 peptide during RF3-mediated recycling of RF1

Michael Graf [1], Paul Huter[1], Cristina Maracci [2], Miroslav Peterek[3], Marina V. Rodnina [2] & Daniel N. Wilson [1]

During translation termination in bacteria, the release factors RF1 and RF2 are recycled from the ribosome by RF3. While high-resolution structures of the individual termination factors on the ribosome exist, direct structural insight into how RF3 mediates dissociation of the decoding RFs has been lacking. Here we have used the Apidaecin 137 peptide to trap RF1 together with RF3 on the ribosome and visualize an ensemble of termination intermediates using cryo-electron microscopy. Binding of RF3 to the ribosome induces small subunit (SSU) rotation and swivelling of the head, yielding intermediate states with shifted P-site tRNAs and RF1 conformations. RF3 does not directly eject RF1 from the ribosome, but rather induces full rotation of the SSU that indirectly dislodges RF1 from its binding site. SSU rotation is coupled to the accommodation of the GTPase domain of RF3 on the large subunit (LSU), thereby promoting GTP hydrolysis and dissociation of RF3 from the ribosome.

[1] Institute for Biochemistry and Molecular Biology, University of Hamburg, Martin-Luther-King-Platz 6, 20146 Hamburg, Germany. [2] Department of Physical Biochemistry, Max Planck Institute for Biophysical Chemistry, Am Fassberg 11, Göttingen 37077, Germany. [3] Central European Institute of Technology (CEITEC), Masaryk University, Kamenice 5, 62500 Brno, Czech Republic. Correspondence and requests for materials should be addressed to D.N.W. (email: daniel.wilson@chemie.uni-hamburg.de)

The termination phase of translation is signalled by the appearance of a stop codon of the mRNA within the A-site of the ribosome. In bacteria, stop codons are recognized by the decoding release factors RF1 and RF2, which facilitate release of the nascent polypeptide chain attached to the P-site tRNA[1–4]. RF1 and RF2 display distinct but overlapping stop codon specificities, such that RF1 decodes UAG and UAA and RF2 decodes UGA and UAA. Both RF1 and RF2 contain a universally conserved GGQ motif that is critical for peptide release[5–9]. Structures of RF1 and RF2 in complex with termination state ribosomes have revealed how conserved residues within the superdomain 2/4 specifically recognize the stop codon on the small subunit (SSU)[10–14]. On the large subunit (LSU), the conserved GGQ motif within domain 3 is located at the peptidyltransferase center (PTC) and facilitates peptidyl-tRNA hydrolysis[10–14]. Following peptidyl-tRNA hydrolysis, the decoding RFs dissociate from the ribosome in a process that is stimulated by the action of a third release factor, the translational GTPase RF3[15,16].

Crystal structures of RF3 confirm structural similarity to other translational GTPases such as EF-Tu[17,18]. Like EF-Tu, RF3 binds to the ribosome with high affinity in the GTP form[19–23]. GTP hydrolysis is not required for the decoding factors to dissociate from the ribosome[21,22], but rather facilitates dissociation of RF3 from the ribosome[22,23]. RF3-GTP binds to ribosomes irrespective of the presence or absence of the decoding release factors, and also interacts with both pre- and post-hydrolysis complexes[21–23]. Biophysical studies indicate that binding of RF3 in the GTP form promotes the conversion of non-rotated RF1- or RF2-bound ribosomes into a rotated state[22,24,25]. Recently, an antimicrobial peptide that binds to the post-hydrolysis ribosome and prevents RF1 dissociation has been reported (Fig. 1a)[22,26]. This peptide, named apidaecin 137 (API), prevents RF1 dissociation even in the presence of RF3[22].

Cryo-EM and X-ray structures exist of RF3-GDP(C/N)P (non-hydrolysable GTP analogues) bound to rotated ribosomes with P/E-hybrid state tRNAs but without the decoding release factors (Fig. 1b)[17,27–29]. Although, the RF3 binding site overlaps with that of other translational GTPases, such as EF-Tu and EF-G, the G-domain of RF3 adopts a distinct orientation on the ribosome[28,29]. Superimposition of the RF3 and RF1/RF2 ribosome structures revealed no overlap in the factor binding sites, suggesting that RF3 indirectly promotes RF1/RF2 dissociation indirectly via inducing ribosomal subunit rotation[14,17,27–30]. A low-resolution (9.7 Å) cryo-EM structure of RF1 and apo-RF3 (no nucleotide form) bound to a non-rotated ribosome has been determined[30]; however, the physiological relevance of this complex remains unclear[21,23]. By contrast, structures of decoding factors on termination state ribosomes in the

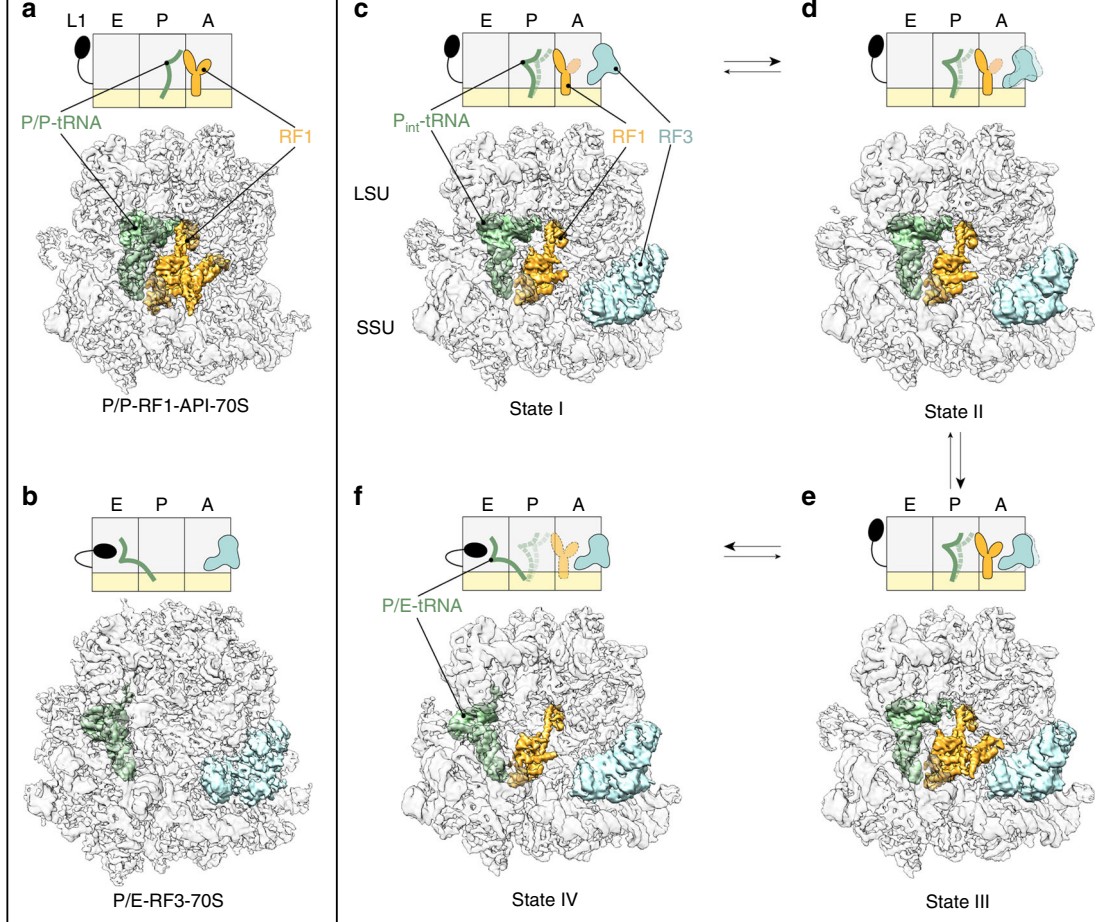

**Fig. 1** Structures of RF1- and RF3-containing termination complexes. **a–f** Schematic representation (above) and electron density (below) for termination complexes containing **a** RF1 (orange) stalled by API on a non-rotated ribosome with a classical P/P-site tRNA (green)[26], **b** RF3 (cyan) trapped by GDPCP on a rotated ribosome with hybrid P/E-site tRNA[28], **c–f** state I–IV with RF1-GAQ (orange), RF3-GDPCP (cyan) bound to **c–e** partially rotated ribosomes with intermediate P-site ($P_{int}$) tRNA (green) or **f** a fully rotated ribosome with a hybrid P/E-site tRNA (green). In the scheme, the SSU and LSU are coloured yellow and grey, respectively, with A-, P- and E-sites and the L1 stalk indicated, whereas flexible regions are indicated by increased transparency. In the map overviews, the electron density for the SSU and LSU has been filtered locally and is shown as a grey transparency so that the ligands can be easily seen within the ribosome

presence of the active GTP-like form of RF3 have so far been lacking.

Here we present an ensemble of structures of tRNA, RF1 and RF3 trapped simultaneously on the ribosome using the termination-specific inhibitor API. The structures reveal that binding of RF3-GDPCP to the complex induces rotation of the SSU relative to the LSU. We do not observe interaction between RF1 and RF3 in any of the structures, suggesting that RF3 mediates dissociation of RF1 indirectly by inducing SSU rotation. SSU rotation also facilitates accommodation of RF3 on the LSU, where the G domain interacts with the sarcin-ricin loop (SRL), which is necessary to stimulate GTP hydrolysis. Thus, RF3-mediated subunit rotation plays a dual role during termination, namely, to dislodge the decoding release factors from the ribosome, but also to facilitate dissociation of RF3 itself.

## Results

**Cryo-EM structures of termination complexes with RF1 and RF3.** In order to visualize both RF1 and RF3 simultaneously on the ribosome, we initially assembled a termination complex in vitro with RF1-GAQ mutant decoding a UAA stop codon in the A-site. This ribosome-tRNA-RF1 complex was then briefly incubated with RF3-GDPCP before being applied to cryo-grids and plunge-frozen. A low-resolution cryo-EM analysis revealed that the termination complex could be sorted into 8 classes, the majority of which contained either non-rotated ribosomes with RF1 but no RF3 or rotated ribosomes bearing RF3 but no RF1

(Supplementary Fig. 1a). The single class that appeared to contain both RF1 and RF3 had strong density for RF3 but poor density for RF1. Binding of RF1 and RF3 to the ribosome thus appeared to be nearly mutually exclusive, suggesting that RF3-GDPCP could recycle RF1-GAQ from the termination complex ribosomes, which is consistent with previous biochemical reports[22,24,25]. To increase the proportion of termination complexes containing both RF1 and RF3 bound simultaneously, we repeated the experiment in the presence of API, which was previously shown to prevent RF1 dissociation even in the presence of RF3-GTP[26]. Because API binds to the exit tunnel and replaces the nascent peptide, by addition of API we selectively stabilized those complexes where the nascent peptide was released despite the use of the RF1 mutant that is slow in catalysing hydrolysis of peptidyl-tRNA[7,9]. Using this complex, cryo-EM data was collected on a Titan Krios transmission electron microscope (TEM) with a Falcon II direct electron detector (DED) and processed with RELION2.1[31]. A total of 525,595 ribosomal particles were sorted into eight distinct ribosomal subpopulations (Supplementary Fig. 1b). The four major subpopulations, states I–IV (15.2–22.4%; 79,975–117,725 particles), all contained P-site tRNA, RF1 and RF3 but were conformational distinct from one another (Fig. 1c–f). States I–IV were refined to average resolutions of 3.8 Å (State I, II) and 3.9 Å (State III and IV) (Supplementary Fig. 1c–g and Table 1). Additionally, four minor subpopulations were present in the dataset, resulting in two additional low-resolution 70S-RF1-RF3 populations (see Methods), vacant 50S subunits (10.3%, 53,850 particles) and RF3

## Table 1 Cryo-EM data collection, refinement and validation statistics

| | State I (EMD 0076, PDB 6GWT) | State II (EMD 0080, PDB 6GXM) | State III (EMD 0081, PDB 6GXN) | State IV (EMD 0082, PDB 6GXO) | RF3-70S (EMD 0083, PDB 6GXP) |
|---|---|---|---|---|---|
| *Data collection* | | | | | |
| Microscope | FEI Titan Krios | FEI Titan Krios | FEI Titan Krios | FEI Titan Krios | FEI Titan Krios |
| Camera | Falcon II | Falcon II | Falcon II | Falcon II | Falcon II |
| Magnification | 131,951 | 131,951 | 131,951 | 131,951 | 131,951 |
| Voltage (kV) | 300 | 300 | 300 | 300 | 300 |
| Electron dose (e−/Å²) | 45.9 | 45.9 | 45.9 | 45.9 | 45.9 |
| Defocus range (μm) | −0.8 to −1.6 | −0.8 to −1.6 | −0.8 to −1.6 | −0.8 to −1.6 | −0.8 to −1.6 |
| Pixel size (Å) | 1.061 | 1.061 | 1.061 | 1.061 | 1.061 |
| Initial particles (no.) | 525,595 | 525,595 | 525,595 | 525,595 | 525,595 |
| Final particles (no.) | 47,512 | 49,415 | 54,142 | 46,814 | 30,535 |
| *Model composition* | | | | | |
| Non-hydrogen atoms | 151,484 | 151,394 | 151,873 | 151,479 | 147,677 |
| Protein residues | 6670 | 6643 | 6742 | 6643 | 6396 |
| RNA bases | 4638 | 4638 | 4637 | 4642 | 4554 |
| *Refinement* | | | | | |
| Resolution (Å) | 3.81 | 3.85 | 3.93 | 3.93 | 4.44 |
| Mask CC | 0.808 | 0.823 | 0.819 | 0.822 | 0.790 |
| Volume CC | 0.798 | 0.813 | 0.810 | 0.812 | 0.775 |
| Map-sharpening *B* factor (Å²) | −125.34 | −125.88 | −129.42 | −126.26 | −134.81 |
| Average *B* factor (Å²) | 187.9 | 204.3 | 221.6 | 206.0 | 354.1 |
| R.m.s. deviations | | | | | |
| Bond lengths (Å) | 0.005 | 0.004 | 0.004 | 0.005 | 0.008 |
| Bond angles (°) | 0.93 | 0.88 | 0.86 | 0.88 | 0.87 |
| *Validation* | | | | | |
| MolProbity score[a] | 1.83 (100th) | 1.84 (100th) | 1.80 (100th) | 1.89 (100th) | 1.79 (100th) |
| Clashscore[b] | 5.92 (100th) | 5.95 (100th) | 5.51 (100th) | 6.90 (100th) | 5.24 (100th) |
| Poor rotamers (%) | 0.20% | 0.24% | 0.20% | 0.17% | 0.27% |
| *Ramachandran plot* | | | | | |
| Favoured (%) | 91.39% | 91.22% | 91.47% | 91.32% | 91.14% |
| Allowed (%) | 8.17% | 8.37% | 8.09% | 8.25% | 8.40% |
| Disallowed (%) | 0.44% | 0.41% | 0.44% | 0.43% | 0.46% |

[a](3.25 Å–4.05 Å)
[b](3 Å–9999 Å)

bound to rotated vacant 70S ribosomes (5.8%, 30,535 particles) (Supplementary Fig. 1b). Since the latter subpopulation did not contain a P/E-tRNA, we believe it represents a state where RF3-GDPCP bound directly to vacant 70S ribosomes, rather than to the ribosome-RF1-GAQ complexes. Local resolution calculations of the RF3-70S complex revealed that while the core of the ribosomal subunits reaches 4.0 Å (Supplementary Fig. 2a, b), there is high conformational flexibility in this state. This is particularly evident in the rotation of SSU relative to the LSU and swivelling of the SSU head, but also in the positioning of the uL1 stalk and RF3 itself (Supplementary Fig. 2a, b). By contrast, states I–IV are more conformationally homogeneous, with local resolutions reaching 3.5 Å within the core of both ribosomal subunits. Flexibility is predominantly observed at the periphery of the ribosome, namely, for the uL1, uL11 and bL12 stalks where local resolutions exceeded 7.5 Å (Supplementary Fig. 2a, b). Local resolution calculations also indicated some conformational flexibility within the ribosome-bound

ligands (Supplementary Fig. 2c). The resolution of the P-site tRNA and RF1 was highest (3.5–4.0 Å) for the regions that interact with the SSU and LSU; whereas, the linking regions were significantly worse (>7.5 Å), such as the elbow of the tRNA, RF1 domain I and the linker between domains II and III of RF1 (Supplementary Fig. 2c). In states I–IV, the local resolution of the ligands was significantly better than that observed in the RF3-70S complex (Supplementary Fig. 2c). Molecular models of states I–IV and the RF3-70S complex were initially generated using rigid-body and domain-wise fitting of the ribosomal subunits, tRNA, RF1 and RF3 crystal structures to the cryo-EM map density, before manual adjustment, refinement and validation (Supplementary Fig. 2d; see Methods; Table 1). All states contained API bound within the ribosomal exit tunnel, where the interaction between Arg17 of API and the Gln235 (Q235 of the GGQ motif) of RF1 (Supplementary Fig. 3a–e) traps RF1 on the ribosome as reported previously[26].

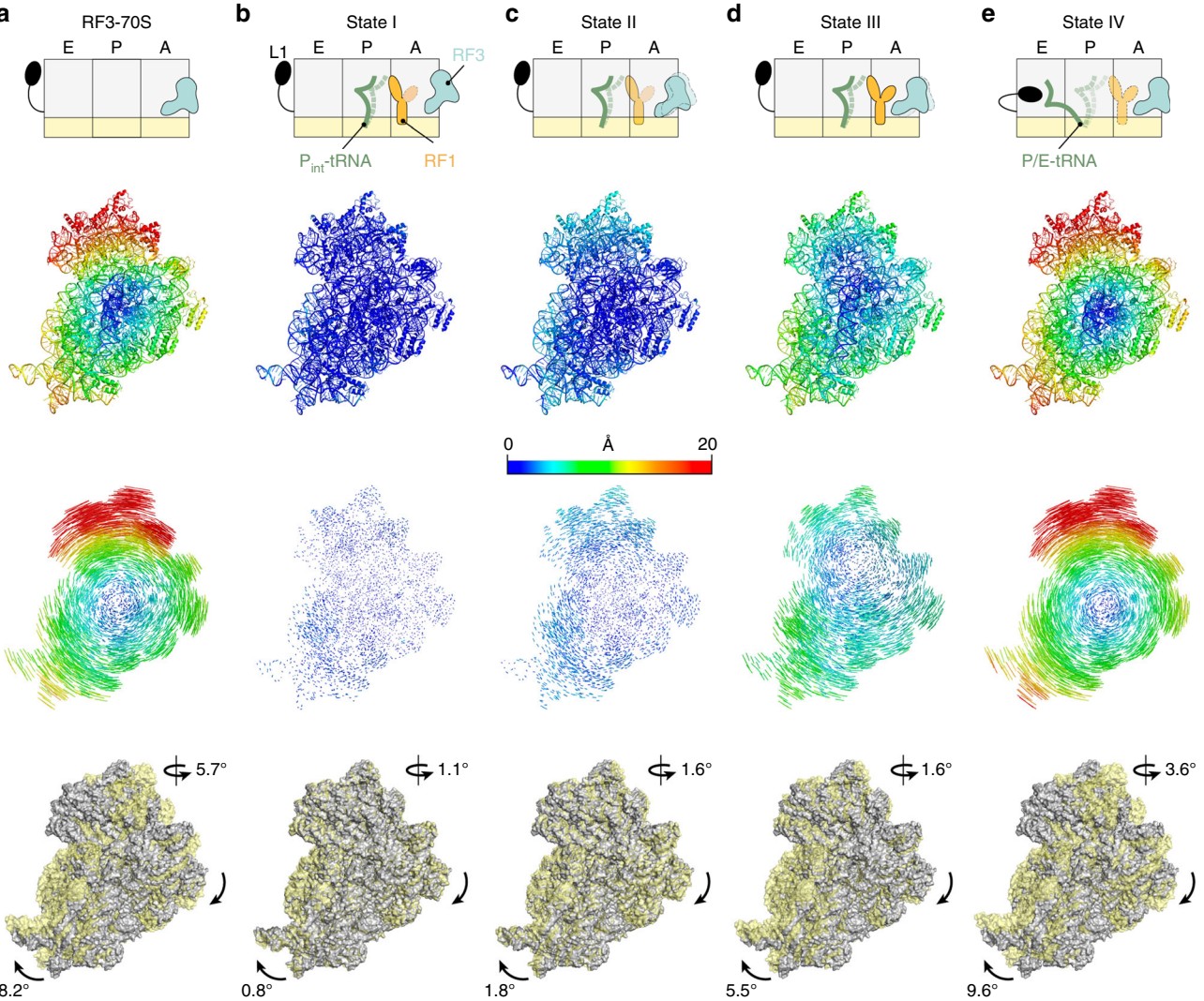

**Fig. 2** Subunit rotation and head swivel observed in RF3-70S and termination state I–IV complexes. **a–e** Schematic representation (upper row) and SSU structures illustrating the degree of rotation relative to non-rotated RF1-API-70S reference structure[26] as shown for **a** RF3-70S, **b** state I, **c** state II, **d** state III and **e** state IV. The distance each atom shifts relative to the reference structure is directly coloured on the SSU (second row), shown as coloured lines connecting the same atoms between the reference and the shifted structure (third row). Superimposition of cryo-EM maps (row four) of shifted SSU (yellow) relative to reference SSU (grey) based on LSU alignments, with degrees of SSU rotation (clockwise as viewed from intersubunit interface) and head swivel indicated

**RF3 induces ribosomal subunit rotation and head swivel.** The major global movements distinguishing states I–IV and the RF3-70S complex are the rotation of the SSU relative to the LSU as well as swivelling of the SSU head relative to the body (Fig. 2a–e and Supplementary Movie 1). The previously reported X-ray crystallography structures of RF3-70S complexes[28,29] revealed SSU rotation of ~10° (clockwise when viewing the intersubunit interface of the SSU) compared to a classical (non-rotated) ribosome, such as the RF1-API-70S complex[26] (Supplementary Table 1). We also observed a similar degree of subunit rotation in the RF3-70S complex (Fig. 2a), whereas states I, II, III and IV displayed a range of intermediate levels of rotation, namely, 0.8°, 1.8°, 5.5° and 9.6°, respectively (Fig. 2b–e and Supplementary Movie 1). In the X-ray crystallography structures of the RF3-70S complexes, the degree of head swivel differed dramatically from one another and was suggested to be dependent on the presence or absence of the hybrid P/E-site tRNA[28,29]. In the absence of the P/E-site tRNA, the head was swivelled ~14° compared to the body[29], whereas only ~3–4° head swivelling was observed when the P/E-site tRNA was present[28] (Supplementary Table 1 and Supplementary Movie 2). In our RF3-70S complex lacking a tRNA (Supplementary Fig. 2), we observed an intermediate level (~6°) of head swivel (Fig. 2a, Supplementary Table 1 and Supplementary Movie 2). However, as noted above, the head is highly dynamic in our RF3-70S complex (Supplementary Fig. 2a, b) and thus the value reflects an average of multiple different swivel conformations of the head. This supports the suggestion that the large degree of head swivel observed in one of the X-ray crystallography structure of the RF3-70S complex is indeed due to the absence of the P/E-site tRNA[29]. Consistently, in states I–IV that contain P- or P/E-site tRNAs, the maximum head swivel observed was ~3.6° (Fig. 2b–e and Supplementary Table 1). Moreover, the degree of head swivel appeared to be loosely correlated to that of subunit rotation, as the degree of swivelling also increased from state I to IV, namely, 1.1° to 3.6°, respectively (Fig. 2b–e). The degree of rotation (~10°) and head swivel (~4–6°) observed here in state IV and RF3-70S complex is similar to that observed previously for translation elongation states with hybrid A/P- and P/E-site tRNAs (Supplementary Table 1)[32–37].

**Remodelled interactions between the P-site tRNA and P-loop.** The SSU rotation and head swivel observed in states I–IV is also accompanied by a corresponding shift of the P-site tRNA (Fig. 3a and Supplementary Movie 1). Compared to the classical P-site tRNA in the RF1-API-70S complex in the absence of RF3, the tRNA is rotated towards the E-site by ~13° in state I–III and by ~34° in state IV (Fig. 3a). For state IV, this generates a hybrid

P/E-tRNA where the CCA-end of the tRNA interacts with the E-site on the LSU, as observed for hybrid P/E-site tRNA during translation elongation[32–37]. By contrast, the intermediate P-site tRNA positions observed in states I–III still have the CCA-end located in the vicinity of the peptidyltransferase center (PTC) of the LSU. A classical P-site tRNA is positioned at the PTC via base-pairing of the C74 and C75 of the CCA-end of the P-site tRNA with nucleotides G2252 and G2251, respectively, of the P-loop (helix H80) of the 23S rRNA (Fig. 3b). By contrast, the rotation of the P-site tRNA observed in states I–III results in a shift of the CCA-end of the P-site tRNA out of the PTC by ~9 Å (Fig. 3c, d). Surprisingly, we observed a register shift in the base-pairing of the CCA-end of the P-site tRNA intermediate with the P-loop nucleotides, such that C74 and C75 were base-paired with G2253 and G2252, respectively. In addition, A73 of the P-site tRNA appeared to flip to establish a non-canonical wobble base-pair with C2254 (Fig. 3c, d). In contrast to the canonical P-site tRNA where the electron density was clearly resolved for the complete CCA-end, no density was visible for the terminal A76 of the P-site tRNA intermediate in states I–III, therefore assignment of the acylation state of the $P_{int}$-tRNA was not possible (Supplementary Fig. 3f–h). We do not believe that the register shift of the CCA-end of the $P_{int}$-tRNA is by two nucleotides, such that A76, C75 and C74 base-pair with G2252, G2253 and C2254, respectively, because the density does not support a purine-purine (A76-G2252) interaction in the first position, nor a pyrimidine-pyrimidine (C74-C2254) interaction at the third position (Supplementary Fig. 3i). Moreover, the density for A73 observed in the P/P-tRNA is not observed in the $P_{int}$-tRNA (Supplementary Fig. 3j–l), which would be expected if C74 interacts with C2254 and prevents A73 from establishing this interaction. Regardless of one or two nucleotide register shift, to our knowledge, such re-base-pairing of the CCA-end of a P-site tRNA with the P-loop has not been observed previously.

**Subunit rotation accommodates RF3 on the LSU.** In the RF3-70S complex and states I–IV, RF3 is observed to rotate as a rigid body together with the SSU (Fig. 4a and Supplementary Movie 1). The overall conformation of RF3 is the same as observed in the previous RF3-70S complexes[28,29], differing from the free RF3 form by the shift of domains 2 and 3 relative to domain 1[17,18,28,29]. Because of the coordinated movement of RF3 with the SSU, the interactions between domain 2 and 3 of RF3 with ribosomal protein uS12 and helices h5 and h15 of the 16S rRNA, as described previously[28,29], are maintained in all states. Relative to the LSU, however, RF3 moves by up to 10–11 Å when comparing states I–IV, bringing the G-domain of RF3 in state IV into

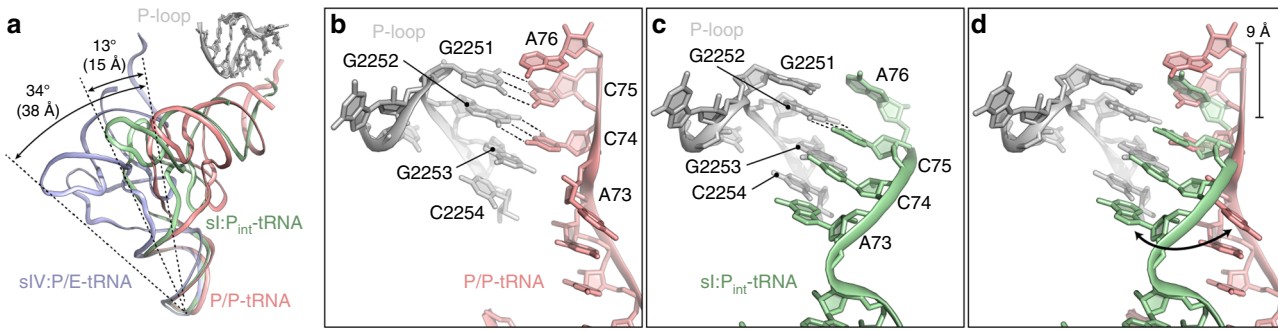

**Fig. 3** Small subunit rotation induces P-site tRNA rotation. **a** Comparison of the relative position of a classical P/P-tRNA (salmon) from the RF1-API-70S complex[26] to $P_{int}$ tRNA (green) conformation observed in state I and the hybrid P/E-site tRNA observed in state IV (slate), with the degree of rotation and distance shifted indicated. The P-loop of the 23S rRNA is shown for reference. **b** The P/P-site tRNA (salmon) in the RF1-API-70S complex[26] and **c** the $P_{int}$-tRNA (green) in state I interact with nucleotides within the P-loop (H80) of the 23S rRNA. Potential hydrogen bonds are indicated with dashed lines. **d** Superimposition of **b** and **c** with arrow indicating the flipped A73 nucleotide

closer proximity of the sarcin-ricin loop (SRL, H95 of the 23S rRNA) (Fig. 4b and Supplementary Fig. 4a). Because the SRL is critical for stimulating the GTPase activity of translation factors[38], this suggests that the SSU rotation is necessary for efficient activation of the GTPase activity of RF3. Evidence for progressive accommodation of translational GTPases on the LSU, as well as GTPase activation by the SRL, has been observed for other translational GTPases, such as IF2[39,40], eEF1A[41,42], SelB[32] and EF-Tu[43] (Supplementary Fig. 4b). However, in these latter cases, the translational GTPases bind to non-rotated ribosomes and accommodation of the GTPase appears to be mediated by SSU domain closure[32,41–43], rather than by rotation as observed here for RF3. Despite the sequence and structural conservation of the G-domain of RF3 with other translation GTPases, the G-domain of RF3 adopts a distinct orientation on the ribosome[28,29]. With respect to the SRL, the G-domain of RF3 is rotated by 24–31° when compared with other translational GTPases, such as IF2, EF-Tu and EF-G (Supplementary Fig. 4c–f). Within the limits of the resolution, the switch II loop conformation of the G-domain of RF3 in states I–IV is consistent with that observed in previous structures of RF3[17,18,28,29], where it interacts with the γ-phosphate of the GDPCP (Fig. 4c). While the switch I loop is poorly ordered in states II–IV, we observed a well-defined conformation in state I, where it interacts with ribosomal proteins uL14 and bL19, but not with the SRL (Fig. 4c). The switch I loop is disordered in many previous structures of RF3[17,18,28], although ordered conformations were previously reported in the *E. coli* RF3-70S structure[29] as well as the *Desulfovibrio vulgaris* RF3 in complex with the alarmone ppGpp[18]. However, they are significantly different from that observed here in state I

(Supplementary Fig. 5a–f). The interaction of the switch I loop conformation of RF3 with uL14 and bL19 is the only direct contact that RF3 establishes with the LSU in state I, and thus may be important for facilitating accommodation of RF3 on the ribosome.

**A dual role for bL12 during translation termination**. The GTPase activity of translational GTPases, such as IF2[44,45], EF-Tu and EF-G[46,47] as well as RF3[44], is stimulated by the ribosomal bL12 stalk, a pentameric complex consisting of four copies of bL12 tethered to the ribosomal protein uL10. In states I–IV, we observe an additional density that we attribute to the C-terminal domain (CTD) of one copy of bL12 interacting with the G' domain of RF3 (Fig. 5a, b), as observed in previous RF3-70S cryo-EM structures[17,30], and consistent with NMR[48] and mutagenesis studies[44]. bL12 was shown to stimulate Pi release from EF-G following hydrolysis of GTP to GDP and Pi, enabling the low affinity GDP conformation of EF-G to be adopted and thereby facilitating the dissociation of EF-G from the ribosome[49]. However, bL12 has also been implicated in promoting binding of translational GTPases, such as EF-G and EF-Tu, to the ribosome[47,50]. Surprisingly, in state III, an extra density is observed that we attributed to a second CTD of bL12, which bridges domain I of RF1 with the ribosomal protein uL11 (Fig. 5b). This extra density can also be seen in the cryo-EM map of the previously reported RF1-API-70S complex (Supplementary Fig. 5g, h)[26], but cannot be seen in states I–II and IV due to the delocalized RF1 domain I. In state III, domain I of RF1 is better resolved, compared to states I and II, due to head swivelling on

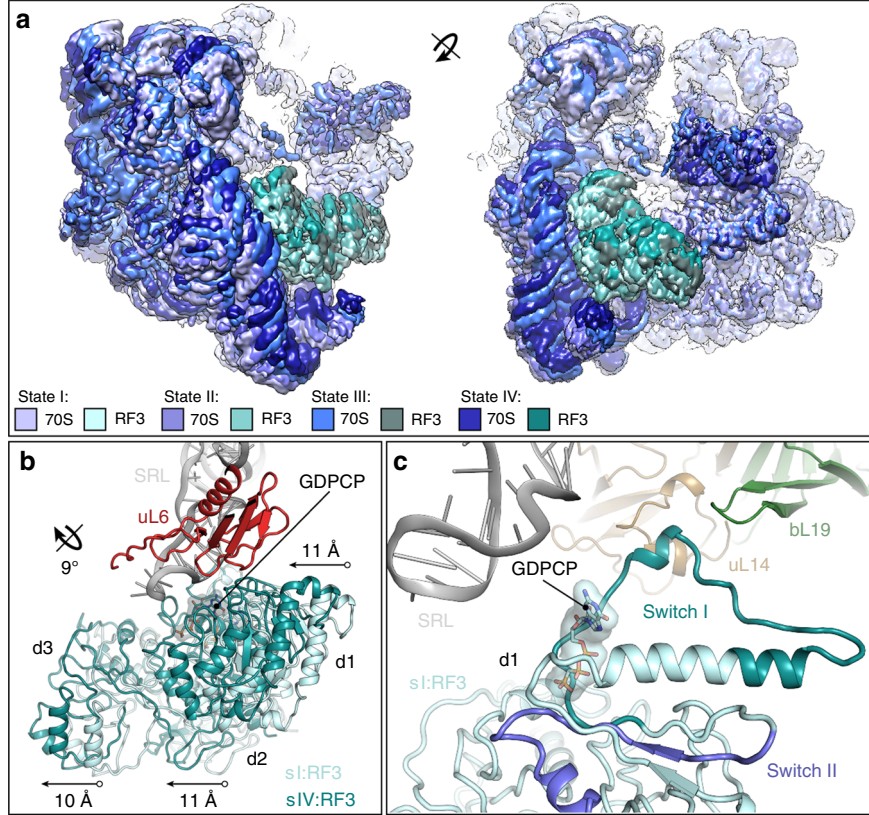

**Fig. 4** Small subunit rotation leads to RF3 accommodation on the large subunit. **a** Two views of the cryo-EM maps of the 70S ribosome (different shades of blue) and RF3 (different shades of green) from states I–IV, illustrating the coupled rotation of the SSU and RF3 relative to the LSU. **b** Comparison of the binding site of RF3 in state I (pale cyan, sI:RF3) and state IV (teal, sIV:RF3) relative to ribosomal protein uL6 (red) and sarcin-ricin loop (SRL, grey). The distance shifted of each domain (d1–d3) of RF3 between states I and IV is indicated. **c** View showing the G domain of RF3 in state I (pale cyan, sI:RF3) with switch I (teal), switch II (slate), GDPCP and proximity to SRL (grey), ribosomal proteins uL14 (tan) and bL19 (green)

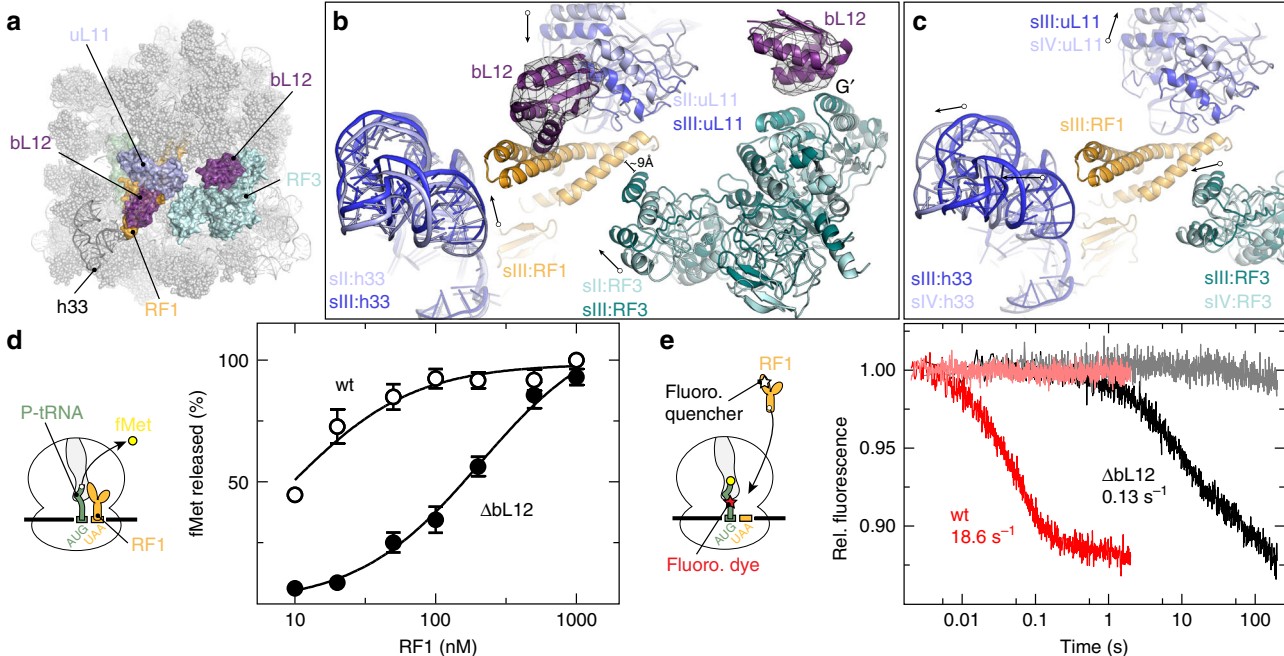

**Fig. 5** Interaction of bL12 with RF1 and RF3 in state III. **a** Overview showing the relative position of RF1 (orange), RF3 (pale cyan), uL11 (light blue), 23S rRNA helix h33 (dark grey) and two copies of the bL12 CTD (purple) on the ribosome (light grey). **b** Comparison of state II position for RF3 (pale cyan), uL11 and h33 (light blue) with state III positions for RF3 (teal), uL11 and helix h33 (dark blue). In state III, RF1 domain I (orange) becomes ordered and density (grey mesh, filtered to 7 Å) for two copies of bL12 CTD (purple) are observed. **c** Comparison of state III positions from (**b**) with state IV positions for RF3 (pale cyan), uL11 and h33 (light blue). **d** Peptide hydrolysis by RF1 in the presence of wild type (wt, open circles) or bL12-depleted (ΔbL12, closed circles) ribosomes. Pre-hydrolysis (PreHC) complexes (0.01 μM) were incubated with increasing concentrations of RF1 for 10 s at 37 °C. Solid lines represent the hyperbolic fit of the experimental points. Error bars represent the standard deviation of the mean for four technical replicates from two independent biological experiments. The apparent affinities of RF1 for wt and ΔbL12 PreHCs are 9 ± 1 and 210 ± 30 nM, respectively. **e** Time courses of RF1-GAQ$_{Qsy9}$ (0.3 μM) binding to PreHC$_{Flu}$ (0.05 μM) prepared with wt (red) or ΔbL12 (black) ribosomes. Buffer controls are shown in salmon and grey, respectively. Traces shown are the average of four to five technical replicates

the SSU and closure of the uL11 stalk base of the LSU (Fig. 5b). By contrast, transition from state III to IV involves additional head swivelling and opening of the uL11 stalk base, which leads to loss of interaction and destabilization of domain I of RF1 (Fig. 5c). Thus, our observations suggest that in addition to stimulating the GTPase activity of RF3, bL12 may also be involved in facilitating the binding of the termination decoding factors to the ribosome. To test this hypothesis we prepared ribosome termination complexes using bL12-depleted (ΔbL12) ribosomes[46] and determined the apparent affinity of RF1 by peptide hydrolysis (Fig. 5d, see Methods). The ΔbL12 ribosomes showed a markedly decreased affinity for RF1, compared to wild-type (wt) ribosomes bearing bL12 (Fig. 5d). We further confirmed this result by measuring the kinetics of RF1 binding via fluorescence resonance energy transfer from a dye-labelled fMet-tRNA$^{fMet}$ and a quencher-labelled RF1-GAQ (RF1-GAQ$_{Qsy9}$) (Fig. 5e)[26]. Binding of RF1-GAQ to the ΔbL12 ribosomes was 150-fold slower than to wt ribosomes, indicating that the contribution of bL12 to RF1 recruitment is large.

**Subunit rotation destabilizes RF1 binding**. Unlike the previous low-resolution (9.7 Å) cryo-EM structure of apo-RF3-RF1-70S complex where contact was reported between RF1 and RF3[30], we do not observe interaction between RF1 and RF3 in any of the structures determined here. The closest distance between the two factors is seen in state III where domain I of RF1 becomes ordered such that α-helix 2 of domain 1 of RF1 comes within 9 Å of RF3 domain 3 (Fig. 5b). Moreover, since RF3 can recycle RF1 variants lacking domain I[51], we conclude that RF3 does not use a direct

steric overlap in binding site to dissociate RF1 from the ribosome. Instead, our results suggest that RF3 dislodges RF1 from its binding site, indirectly, by inducing SSU rotation, as postulated previously[17,28,29]. The density for RF1 remains well-resolved through-out states I–III, indicating that RF1 is stably bound to the ribosome despite the increase in rotation (0.8–5.5°) of the SSU (Fig. 6a). By contrast, the density for RF1 is poorly resolved in state IV, indicating that RF1 becomes destabilized from its binding site on the ribosome (Fig. 6a, b). Increased flexibility of RF1 in state IV is also supported by local resolution calculations (Supplementary Fig. 2c). Transition from state III to IV involves further clockwise rotation of the SSU (from 5.5° to 9.6°), which results in a shift of domain 2/4 of RF1 by 4 Å compared to state III (Fig. 6a, c). Because domain 3 of RF1 remains static at the PTC of the LSU, the movement can be described by a rotation of 6° that is accommodated by the long flexible linkers connecting domain 2/4 with domain 3 of RF1 (Fig. 6c). We believe that it was possible to capture the RF1 conformation in state IV only because the complex was formed in the presence of API, which prevented the complete dissociation of RF1 from the rotated ribosome. We note that transition from state III to IV also encompasses additional head swivel (from 1.6° to 3.6°) as well as opening of the bL12 stalk base (Fig. 5c), both of which destabilize domain 1 of RF1 and may facilitate dissociation of RF1 from the ribosome. Additionally, the formation of a hybrid P/E-tRNA due to the fully rotated SSU in state IV also leads to loss of interaction of RF1 with the P/P-tRNA, which may also contribute to destabilization of RF1 binding. Specifically, the interactions between Glu155 and His156 in domain 2/4 and Arg261 in domain 3 of RF1 with the ASL and CCA-end of a P/ P-tRNA, respectively, are lost upon P/E-tRNA in state IV (Fig. 6d, e).

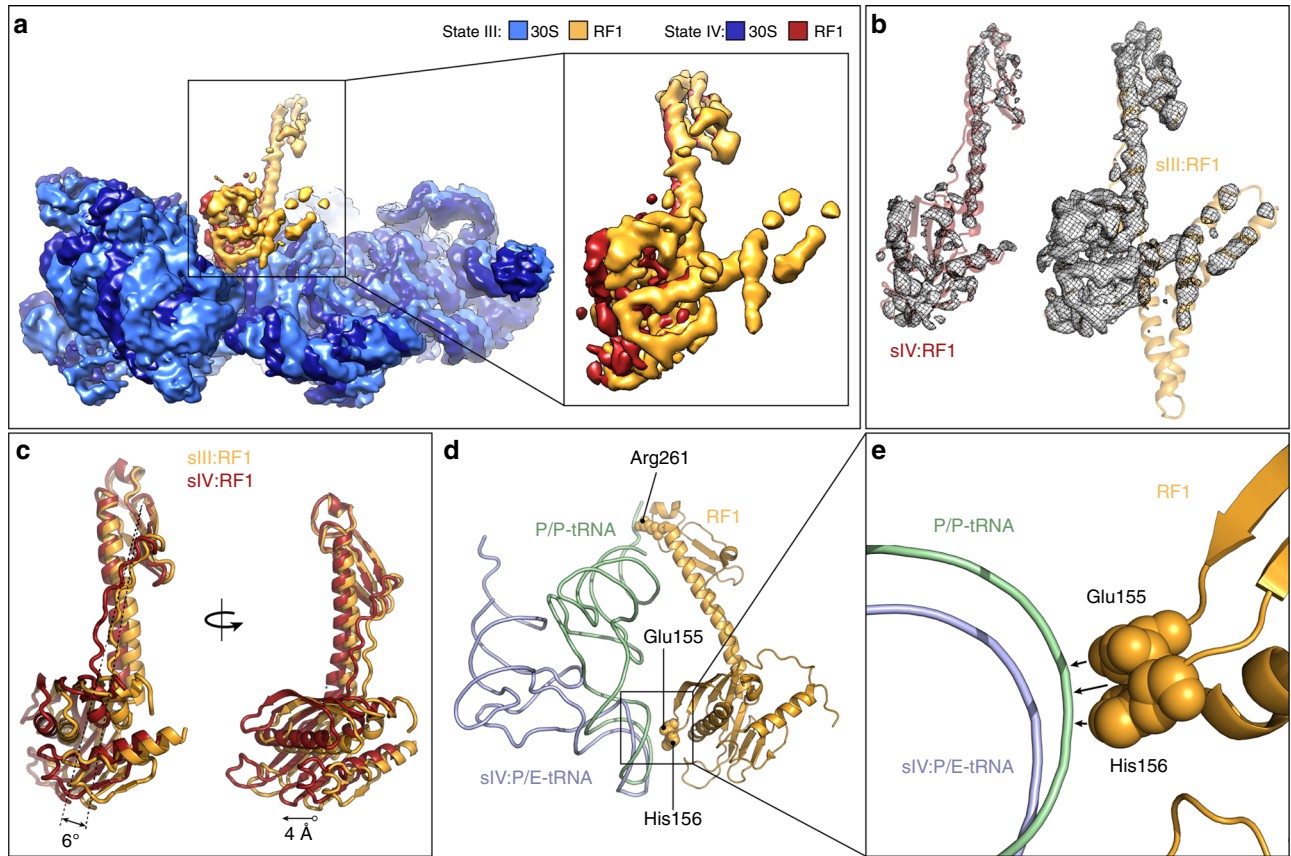

**Fig. 6** RF3-induced subunit rotation destabilizes RF1 binding. **a** Cryo-EM map of SSU (light blue) and RF1 (orange) from state III compared with SSU (dark blue) and RF1 (red) from state IV. **b** Isolated cryo-EM electron densities (grey mesh) with molecular models for RF1 from state III (orange) and state IV (red) shown at the same contour level based on comparison with the SSU density. **c** Domain 2/4 of RF1 from state III (sIII:RF1, orange) is rotated by 6° and shifted by 4 Å compared to RF1 from state IV (sIV:RF1, red). **d**, **e** Contacts (arrowed) between RF1 (orange) and P/P-tRNA (green) are lost upon formation of the hybrid P/E-tRNA (light blue). Amino acids of RF1 that contact P/P-tRNA are shown as spheres. **e** Zoom of **d** showing the presence or absence of RF1 contacts with the ASL of P/P- or P/E-tRNA, respectively

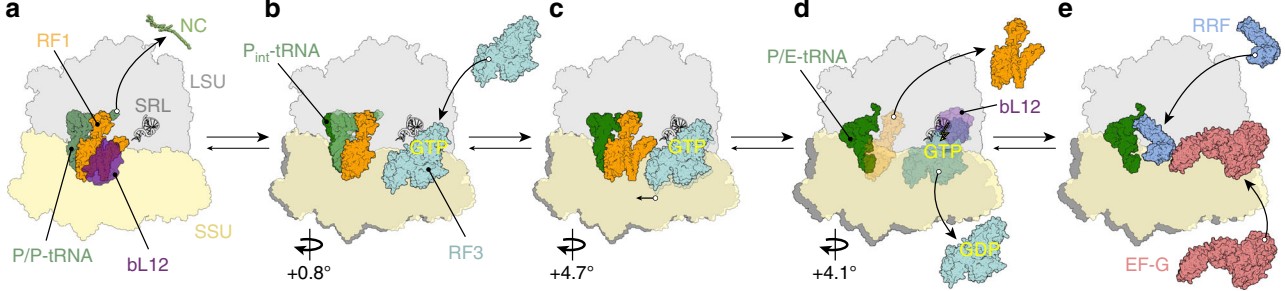

**Fig. 7** Model for RF3-mediated dissociation of RF1 from the ribosome. **a** Binding of decoding release factors, such as RF1 (orange) to the non-rotated termination state ribosome, leads to release of the nascent polypeptide chain (NC) from the P-site tRNA (green). Binding of RF1 to the ribosome is facilitated by interaction of bL12 CTD (purple) with domain 1 of RF1. **b** RF3 (cyan) in the GTP conformation binds to the ribosome-RF1 complex. RF3 binding induces a slight rotation of the SSU that promotes formation of the partially rotated P-site tRNA conformation ($P_{int}$-tRNA). **c** Additional SSU rotation and head swivel stabilizes domain I of RF1 and induces a closed conformation of uL11 (Fig. 5b, c). **d** Further SSU rotation leads to destabilization of RF1, promoting its dissociation from the ribosome, as well as accommodation of RF3 on the LSU in close proximity to the sarcin-ricin loop (SRL), thus facilitating hydrolysis of GTP to GDP and dissociation of RF3-GDP from the ribosome. **e** The rotated ribosome with a P/E-tRNA is recognized by the ribosome recycling factor (RRF) and EF-G, which recycle the post-termination complex for the next round of translation

## Discussion

Based on our ensemble of termination intermediates (states I–IV), as well as the available literature, we suggest a revised model for RF3-mediated dissociation of RF1 during translation termination (Fig. 7a–e). A stop codon in the A site is recognized by the decoding factors RF1 (or RF2), which catalyse hydrolysis of the peptidyl-tRNA (Fig. 7a). The observed interaction between the CTD of one copy of bL12 and domain 1 of RF1 in the RF1-API-70S complex (Fig. 7a), as well as the RF1-API-RF3-70S complex (state I), is supported by our experimental findings

(Fig. 5d, e) showing that bL12 plays an important role in delivery of RF1 to the ribosome, as reported previously for EF-Tu and EF-G[46,47]. To date, all reported structures of decoding factors RF1 (and RF2) were bound to ribosomes with non-rotated conformations[14,52,53], including the apo-RF3-RF1-70S complex[30]. By contrast, we observe that binding of RF3 to the ribosome in states I–IV that contain RF1 induces rotation of the SSU relative to the LSU (Fig. 7b–e). While we have ordered states I–IV based on the degree of subunit rotation, we acknowledge that the exact biological sequence of states cannot be ascertained from our study. Nevertheless, we believe that this order produces a logical sequence of events that provide a working model for decoding factor recycling by RF3. In state I, we observe that even a small degree (0.8°) of subunit rotation induced by RF3 results in a shifted position of the P-site tRNA, such that it partially rotates out of the PTC in the direction of the E-site (Fig. 7b). The resulting intermediate P-site tRNA ($P_{int}$-tRNA) displays register shift in the base-pairing between the CCA-end of the tRNA and the P-loop of the PTC (Fig. 3). Moreover, we observe ordering of the switch I loop in the G domain of RF3 (Fig. 4c), which establishes the sole interaction with the LSU and may therefore be important for accommodation of RF3 on the ribosome. Further subunit rotation (+4.7°) as well as the head swivel observed in state III leads to a stabilization of domain 1 of RF1 and further accommodation of RF3 on the LSU (Fig. 7c). In state III, domain I of RF1 comes within 9 Å of RF3 (Fig. 5b), which is the closest distance between the two factors in any of the structures reported here. The absence of an observed interaction between RF1 and RF3 in states I–IV suggests that RF3 must indirectly induce RF1 dissociation from the ribosome. Indeed, in state IV, we observe that further rotation (+4.1°) of the SSU leads to a destabilization in the binding of RF1 (Fig. 7d). We note that transition from states III to IV also encompasses additional head swivelling and movement of the uL11 away from RF1, as well as loss of interaction with the P/E-tRNA, which may also contribute to the destabilization of RF1 binding (Fig. 5c). The large degree (9.6°) of SSU rotation observed in state IV brings the G domain of RF3 into close proximity of the SRL on the LSU (Fig. 7d). Thus, we predict that subunit rotation is necessary to stimulate the GTPase activity of RF3 and thereby facilitate dissociation of the low affinity RF3-GDP from the ribosome (Fig. 7d). As reported previously, we also observe interaction of the CTD of bL12 with the G' domain of RF3 in states I–IV, suggesting that bL12 may also play a role in GTPase activation and dissociation of RF3 (Fig. 7d). Because RF1 and RF3 are trapped on the ribosome with API and GDPCP in our structures, we cannot distinguish the order of dissociation of RF1 and RF3. Nevertheless, our structures suggest that RF1 and RF3 dissociation are both coupled to full rotation of the SSU, and are therefore likely to occur within a very similar timeframe, as reported recently using biophysical assays[22,54]. We note the product remaining after the action of RF1 and RF3 is a rotated ribosome complex with a hybrid P/E-site tRNA, which is the exact substrate for the next phase of translation, namely, ribosome recycling via the binding of RRF and EF-G[55–60] (Fig. 7e).

## Methods

**Preparation of the ribosomal complex.** Ribosomes from the *E. coli* strain MRE600, initiation factors, fMet-tRNA^fMet and its fluorescein-labelled derivative were prepared as described[61,62]. bL12-depleted ribosomes were prepared by NH₄Cl and ethanol treatment as described[46]. The ribosome complexes were assembled on the synthetic 'start-stop' mRNA (5′-GGCAAGGAGGUAAAUAAAUG**UAA**ACGAUU-3′) as follows: 70S (1 μM), initiation factors IF1, IF2 (2 μM) and IF3 (1.5 μM), start-stop mRNA (3 μM) and f[³H]Met-tRNA^fMet (1.5 μM) were incubated in buffer A (30 mM HEPES, pH 7.4, 70 mM NH₄Cl, 5 mM MgCl₂, 30 mM KCl) in the presence of GTP (1 mM) for 30 min at 37 °C and purified through sucrose cushion as described[23]. The

ribosome pellets were resuspended in buffer A, flash frozen in liquid nitrogen, and stored at −80 °C.

***E. coli* strain and growth conditions.** *E. coli* BL21 strain was used for the expression of RF1-GAQ and RF3. Cells were grown in LB medium supplemented with the required antibiotic, and expression was induced by addition of 0.5 mM IPTG.

**Purification of peptide chain release factors.** RF1, RF1-GAQ (RF1-G234A), RF3 and the single-cysteine RF1-GAQ were expressed and purified by affinity chromatography as described[22,26]. RF1-GAQ was labelled with Qsy9[26].

**Cryo-grid preparation for the 70S-tRNA-RF1-RF3 complex.** All following steps were performed in buffer A (30 mM HEPES, pH 7.4, 70 mM NH₄Cl, 5 mM MgCl₂, 30 mM KCl). For grid preparation, 5 OD A₂₆₀ per ml of ribosomes were used. RF3 was initially incubated with 1 mM of GDPCP at 37 °C for 15 min. Subsequently, the ribosome-tRNA complexes were incubated with a 2.5× excess of the RF1-GAQ mutant, which is 3300-fold slower than the wild-type RF1[7,9], and with or without 50 μM API (NovoPro Biosciences Inc.) for 1 min at room temperature. Afterwards a 7.5× excess of RF3-GDPCP over 70S ribosomes was added to the RF1-GAQ containing ribosome complexes and kept on ice for <5 min before cryo-grid preparation. All samples were applied to 2 nm precoated Quantifoil R3/3 holey carbon supported grids and vitrified using a Vitrobot Mark IV (FEI, Netherlands).

**Cryo-electron microscopy and single-particle reconstruction.** The low-resolution data collection of the 70S-tRNA-RF1-RF3 complex, which was prepared in the absence of API, was conducted using a Tecnai G2 Spirit (FEI) transmission electron microscope (TEM) equipped with a TemCam-F816 camera (TVIPS) at 120 kV using a pixel size of 2.85 Å. The high-resolution data collection of the 70S-tRNA-RF1-RF3 complex, which was prepared in the presence of API, was performed using an FEI Titan Krios TEM equipped with a Falcon II (FEI) direct electron detector at 300 kV using a pixel size of 1.061 Å and an under-defocus range of −0.8 to −1.6 μm resulting in a total number of 5670 micrographs. Each micrograph was recorded as a series of 16 frames (2.7 e⁻ per Å² pre-exposure; 2.7 e⁻ per Å² dose per frame). All frames (accumulated dose of 45.9 e⁻ per Å²) were aligned using the Unblur software[63], and power spectra, defocus values, astigmatism and estimation of micrograph resolution were determined by GCTF[64]. Micrographs showing Thon rings beyond 4.0 Å resolution were further used. Automatic particle picking was performed using Gautomatch (http://www.mrc-lmb.cam.ac.uk/kzhang/), and single particles were processed using the Relion2.1 software package[31]. Initial 2D classification/alignment was performed with 703,379 particles. Subsequently, promising 2D classes with a total number of 525,595 ribosomal particles were selected and subjected to 3D refinement using an *E. coli*-70S ribosome as a reference structure. Initial alignment and subsequent 3D classification was performed using three times decimated data. The initially refined particles were 3D classified into 8 classes. Class 1–4 were refined again and subjected to another 3D classification. Sorting of class 1–4 resulted in 4, 4, 3 or 3 additional subclasses, respectively. The most stable sub-class of each 3D classification was then 3D-refined. To further increase the resolution of RF3, we applied a focussed mask on RF3. For 3D classification, the same reference was used as for the 3D refinement. The maximum resolution was observed for state I and state II, extending to <3.9 Å (FSC₀.₁₄₃) (Supplementary Fig. 1). The local resolution of the final maps was computed using ResMap[65] (Supplementary Fig. 2). The final maps were sharpened by dividing the maps by the modulation transfer function of the detector and by applying an automatically determined negative *B* factor to the maps using Relion2.1[31]. For model building the final maps were locally filtered using the SPHIRE cryo-EM software suite[66]. Resolution was estimated using the "gold standard" criterion (FSC = 0.143)[67,68].

**Molecular modelling and map-docking procedures.** The molecular models for the ribosome were based on the *E. coli*-70S-EF-Tu structure (PDB: 5AFI)[69]. The models of RF3 and GDPCP are based on the structure of *E. coli* RF3-GDPNP bound to *Thermus thermophilus* 70S (PDB: 4V85)[29]. The structure of RF1-GAQ and API is based on the recently published *E. coli*-70S-API-RF1 structure (PDB: 5O2R)[26]. The tRNA^fMet in the classical state is derived from an *E. coli* 70S initiation complex containing the ribosomal rescue factor ArfA (PDB: 5U9F)[70]. The tRNA^fMet in the P/E hybrid state is based on the hybrid state tRNA^fMet from the *T. thermophilus* RF3-GDPCP-70S structure (PDB: 4V8O)[28]. The rRNA domains and proteins were rigid-body fitted into the respective EM-map using Chimera[71]. The models were manually adjusted and refined using Coot[72]. Due to the lack of density, domain I of RF1-GAQ was not modelled for states I–II and IV, whereas it was possible to generate a poly-Ala model of domain I of RF1-GAQ for state III. The complete atomic model of the respective complexes were refined into the locally filtered maps using phenix.real_space_refine with secondary structure restraints calculated by PHENIX 1.13[73]. Cross-validation against overfitting was performed as described elsewhere[74]. The statistics of the refined model were obtained using MolProbity[75] (Table 1).

**Calculation of rotation angles**. Rotation angles were calculated using PyMol Molecular Graphics Systems (Version 1.8 Schrödinger) (Supplementary Table 1). The body/platform (including h44) rotation was calculated relatively to the 50S ribosomal subunit. Accordingly, for body/platform rotations the molecular models were aligned based on the 23S rRNA using state I (this study) as reference. The head swivel was calculated relatively to the 30S body/platform. In order to get comparable values for head swivelling, the compared molecular models were aligned based on the 16S rRNA of the body/platform using state I (this study) as reference.

**RMSD and vector calculations**. Root mean square deviation (RMSD) values were calculated between related alpha-carbon (protein) or phosphate atoms (rRNA) using PyMol Molecular Graphics Systems (Version 1.8 Schrödinger). The reported RF1-API-70S structure[26] was used as reference. All compared models were aligned to the 50S subunit of state I (this study). The 30S protein or rRNA residues of the reference structure were coloured according to the determined RMSD of each atom. Vectors were calculated between shifted alpha-carbon and phosphate atoms using the same reference[26]. The vectors were coloured based on their length (distance between atoms).

**Peptide hydrolysis**. Ribosome termination complexes (10 nM) prepared with wild type or ΔbL12 ribosomes were mixed with increasing concentrations of RF1 for 10 s at 37 °C in buffer A. Reactions were quenched with a solution containing TCA (10%) and ethanol (50%). After centrifugation (30 min, 16,000×g) the amount of released f[$^3$H]Met in the supernatant was quantified by radioactive counting.

**Rapid kinetics**. Rapid kinetic experiments were performed on an SX-20MV stopped-flow apparatus (Applied Photophysics, Leatherhead, UK), by rapidly mixing equal volumes (60 µl) of reactants at 37 °C in buffer A. Binding of RF1-GAQ was monitored by mixing termination complex labelled with fluorescein (50 nM) with RF1-GAQ$_{QSY9}$ (300 nM). Fluorescein was excited at 470 nm and monitored after passing a KV500 filter (Schott, Mainz, Germany).

**Figure preparation**. All figures showing electron densities and atomic models were generated using UCSF Chimera[71] and PyMol Molecular Graphics Systems (Version 1.8 Schrödinger).

**Data availability**. The data that support the findings of this study are available from the corresponding author upon request. The atomic coordinates and/or the associated maps have been deposited in the PDB and/or EMDB with the accession codes EMD: 0076/PDB: 6GWT (State I), EMD: 0080/PDB: 6GXM (State II), EMD: 0081/PDB: 6GXN (State III), EMD: 0082/PDB: 6GXO (State IV) and EMD: 0083/PDB: 6GXP (RF3-70S).

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

## Acknowledgements

This work has been supported by iNEXT, project number 2992, funded by the Horizon 2020 programme of the European Union. This article reflects only the author's view and the European Commission is not responsible for any use that may be made of the information it contains. CIISB research infrastructure project LM2015043 funded by MEYS CR is gratefully acknowledged for the financial support of the measurements at the CF Cryo-electron Microscopy and Tomography CEITEC MU. This research was supported by grants of the Forschergruppe FOR1805 (to D.N.W. and M.V.R) and WI3285/6-1 (to D.N.W.) from the Deutsche Forschungsgemeinschaft (DFG).

## Author contributions

D.N.W and M.V.R. designed the study, M.G. and C.M. prepared the cryo-EM sample, M.P. collected the cryo-EM data, M.G. processed the cryo-EM data, M.G. and P.H. built and refined the molecular models, M.G., P.H. and D.N.W. analyzed the cryo-EM data. C. M. performed bL12 experiments. M.G. prepared the figures. M.G. and D.N.W. wrote the paper with help from C.M. and M.V.R.

## Additional information

**Competing interests:** The authors declare no competing interests.

