## [Peer Review File · Nature Communications]

Reviewers' comments:

Reviewer #1 (Remarks to the Author):

The role of RF3 in the dissociation of the release factors RF1 and RF2 is unclear. Graf et al. report the cryo-EM structures of RF1 and RF3 bound simultaneously to the E. coli ribosome. To obtain these structures, they have to use both an antimicrobial peptide, API, and a non-hydrolyzable analog of GTP. The authors were unable to obtain a structure with RF1 and RF3 bound simultaneously in the absence of API. This raises the question as to whether the complexes that have been stabilized are physiologically relevant, or an artifact of the stabilizing methods used. Notably, the complex is different from that obtained by Joachim Frank's lab (Elife 2, e00411), who observed a physical contact between RF1 and a reportedly nucleotide-free RF3.

- 1) Is the title factually correct, given that RF3 cannot recycle RF1 in the presence of API?
- 2) It should be made clear in the abstract exactly how this complex was trapped.
- 3) The authors do not show that RF3 “mediates dissociation of RF1 indirectly by inducing SSU rotation” – the structures simply suggest that this might be possible. The language in the manuscript should be revised carefully.
- 4) FRET experiments should be considered to show a correlation between RF1 dissociation and RF3-induced/stabilized subunit rotation. This would strengthen the proposed model.
- 5) The Frank group made RF1 mutants at the interface between RF1 and RF3 that were shown to affect RF3-mediated recycling of RF1, supporting their conclusions of a physical interaction (Elife 2, e00411). These mutants should be discussed in the framework of the structures presented here – is there an alternative explanation as to why the mutations affected RF3-mediated recycling of RF1?

Reviewer #2 (Remarks to the Author):

The molecular mechanism of translation termination in bacteria requires release factors RF1, RF2 that recognize a stop codon in the A-site and facilitate peptidyl-tRNA hydrolysis. The

GTPase RF3 then releases these factors from the ribosome to allow a new translation cycle to restart. Whereas the action of RF1 and RF2 has been characterized in detail, the process of their RF3-mediated dissociation from the ribosome remained elusive, and only 9.7 Å structure of RF1 and apo-RF3 (no nucleotide form) bound to a non-rotated ribosome has been determined. In the current manuscript, Wilson and colleagues aim to describe how RF3 promotes RF1 dissociation using cryo-EM.

The authors first reconstituted a full termination complex with RF1 and then added RF3-GDPCP before subjecting it to cryo-EM. Since it resulted in a low RF1-RF3 occupancy, they cleverly supplemented the reaction mixture with API that prevents RF1 dissociation through replacing the nascent peptide in the tunnel. The consequent cryo-EM data analysis shows four distinct states resolved at 3.8-3.9 Å resolution, with RF1 and RF3 adopting different conformations. The crystal structures of tRNA, RF1 and RF3 were then fitted to the density map to interpret the data.

The series of structures show that binding of RF3-GDPCP induces rotation of the small subunit that results in RF1 dissociation. No direct interactions between RF3 and RF1 are observed. The rotation facilitates interaction between the RF3 G-domain and sarcin-ricin loop that stimulates the GTPase activity. Furthermore the rotation also leads to a shift of the P-site tRNA. Following the action of RF3, the ribosome is in the exact position to accommodate the binding of RRF and EF-G for a restart. An interesting observation is that in addition to the C-terminal domain of one copy of L7/L12, there is an extra density attributed to a second copy bridging domain I of RF1 with L11. Authors then used Δ L7/L12 strain to prove that the stalk is indeed involved in facilitating the binding of termination factors.

Overall, the study provides logical sequence of events that can serve as a working model for the recycling of release factors. A potential point of criticism is that essentially similar mechanism of RF1 release was implied before, however in my opinion the paper provides clear experimental evidence and the first direct observation of how RF3 acts on the ribosome to release RF1, which is of a high scientific value to the field of translation. The work is put well in the context, and as far as I can judge (my field is mt-translation) the previous literature is described well. Figures illustrate the reported results in a clear manner, and schematic insets are very useful for understanding, as well as the supplementary movies.

Minor comments:

- Figure 3: indicating hydrogen bonds at this resolution is ambitious; perhaps authors would consider adding the experimental density here as well.
- For the cryo-EM part, it's a good practice to include a representing micrograph and 2D class averages.
- Please use the updated nomenclature for ribosomal proteins (Curr opin in struct biol 24: 165-

169).

Reviewer #3 (Remarks to the Author):

The manuscript by Graf et al. addresses the molecular mechanism of translation termination in prokaryotes that involves class-I release factors RF1 and RF2 and the class-II release factor RF3, which is a ribosomal GTPase. This study addresses a longstanding question of whether RF1/2 and RF3 co-exist bound to the ribosome, and how release from the ribosome is triggered mechanistically, a fundamental scientific question that several laboratories including ours tried to address in the past.

Using a novel approach with a peptide blocking RF1 dissociation (published recently by a part of the co-authors of this paper, the peptide replaces the nascent peptide in the exit tunnel), a GAQ mutant of RF1 that slows down RF1-dissociation and combining that with GDPCP to block RF3, it became possible to reconstitute a more stable complex than previously without the inhibitor peptide. The structure of various sub-populations was determined by cryo-EM involving significant particle sorting and reached ~ 3.8 Å for the best resolved complex.

This study indeed gives interesting insights into the mechanism of translation termination. In particular, it provides the first structure of a ribosome with both RF1 and RF3 factors bound in the presence of a (non-hydrolysable) GTP analogue. Comparison of the different structures reveals ribosomal subunit rotations with distinct tRNA, RF1 and RF3 positions and conformations, revealing also tRNA translocation. The structure also suggests the presence of an unusual, shifted base-pair formation between the CCA-end of the tRNA and 23S rRNA P-loop of the peptidyl-transferase centre. Other interesting findings include the observation of the ordered switch I loop in the RF3 GTPase and the role of the L7/L12 proteins that contribute to the binding of RF1 (as shown by a complementary functional analysis using L7/L12 deletion mutants), in a manner similar to that observed previously for EF-Tu and EF-G. The fact that RF1 and RF3 co-exist on the ribosome but do not seem to interact directly suggest that the mechanism of RF1 operates indirectly through the observed conformational changes of the ribosome and the 30S head swivel movement in particular.

Overall, this study appears technically well sound and allows making interesting conclusions. It would however be good to clarify a few points. For example, as the resolution of the maps is 3.8 Å for the best resolved complex, is the base pairing between the CCA-end of the tRNA and the P-loop resolved locally well enough to discuss it to a rather detailed extent? Is the flipping of base A73 resolved in the two complexes that are being compared? Maybe some additional picture of the structure including the cryo-EM map could be helpful, especially to ascertain the proposed register shift of the interaction between the CCA-end nucleotides and the P-loop

nucleotides. Along the same lines, the overall structure description seems limited to domains of the factors, is the resolution not sufficient to describe more details? Possibly using cryo-EM image processing techniques like focused refinements (e.g. see series of publications in *Curr Opin Struct Biol* 2017, von Loeffelholz et al., 2017 and references cited therein) could be helpful here and help interpret the cryo-EM maps more easily or in more detail.

With these new data in hand it would now be interesting to discuss some aspects in more detail, considering for example several concepts regarding RF3 function that we described in our early work when we studied the first structure of an RF3-bound ribosome (Klaholz et al., *Nature* 2004) using for the first time particle sorting and revealing 2 different structures and RF3 conformations in equilibrium with each other, for example: the existence of pre- and post-translocated states (tRNA transition from the P- to the E-site plus the associated L1 movement), the RF3-dependent rotation of the ribosomal subunits, the possibility of co-existence of class-I and class-II release factors, etc. (see also our *TIBS* review in 2011 on translation termination, Figure 6), the fact that the G/G' domains move from an inactive RF3 conformation to an activate one towards the α -sarcin-ricin loop where GTP-hydrolysis is induced etc. Now that this can be analysed at much higher resolution it would be an opportunity to re-discuss some of these aspects. In this context it is worth mentioning that we also tried analysing the structure of a ribosome complex with RF2 and RF3 simultaneously bound, but even with structure sorting at the time it would give either RF2 or RF3 bound separately (unpublished), as also found in the present work described in the first part of the paper for RF1- and RF3-bound ribosome subpopulations. Therefore, the API inhibitor peptide is really a good and elegant approach to tackle a difficult problem and stabilize the otherwise transient complex that finally allowed addressing a longstanding question. This study will therefore be of general interest to the field.

Detailed points:

- cryo-EM maps and atomic coordinates should be provided via PDB and EMD databases.
- page 3, 3. line from bottom: typo, should say: "have so far been lacking".
- page 3, same paragraph: the concept that "Superposition of the RF3 and RF1/RF2 ribosome structures revealed no overlap of the factor binding sites" was shown in our early work already (reference 27).
- page 3, last line: "binding of RF3... induces rotation of the SSU relative to the LSU" was seen in previous work also (ref. 27), this could now be discussed in the final discussion part also, as suggested above.
- page 5: maybe specify how slow the RF1 mutant is compared to wild type RF1; this is particularly relevant for this study because if there was no release of the nascent peptide the inhibitor peptide would probably not be active. Apparently it is still fast enough to release the peptide because the incubation with the RF1 mutant was only 1 min as described in the methods section.

- page 6: linking regions of RF1 are at low resolution, maybe cryo-EM focused refinements after particle sorting could help?
- page 7: does the register shift and re-base-pairing imply a bent of the tRNA acceptor stem?
Last phrase of this paragraph: specify that re-base-pairing also comprises a register shift, it does not re-base-pair with the same residues.
- page 8: overall conformation of RF3 as described in references 28,29: is it also similar to one or the other sorted state of RF3 in ref. 27?
- page 8, middle: translational GTPases and activation by SRL, “such as eEF1A, SelB and EF-Tu”: true for IF2 also (e.g. Myasnikov et al., NSMB 2005; Sprink et al., Sci Adv. 2016).
- page 9, role of L7/L12 for the activity of ribosomal GTPases: true for IF2 also (e.g. Carlson et al., FEBS 2017; Ge et al., PNAS 2018).
- page 10: loss of interaction of RF1 with the P/P-tRNA: this interaction is anyway not that strong.
- page 10, bottom: “...all reported structures of decoding factors RF1 and RF2 were bound to ribosomes with non-rotated conformation (ref. 14)”: this is true also for the 2 structures published by Rawat et al. and Klaholz et al., Nature 2003.
- page 11, reference 48 suggests that RF1 leaves the ribosome after RF3, is this possible with the mechanism described here?
- Fig. 1C: add label for RF1.

Reviewer #1 (Remarks to the Author):

The role of RF3 in the dissociation of the release factors RF1 and RF2 is unclear. Graf et al. report the cryo-EM structures of RF1 and RF3 bound simultaneously to the E. coli ribosome. To obtain these structures, they have to use both an antimicrobial peptide, API, and a non-hydrolyzable analog of GTP. The authors were unable to obtain a structure with RF1 and RF3 bound simultaneously in the absence of API. This raises the question as to whether the complexes that have been stabilized are physiologically relevant, or an artifact of the stabilizing methods used. Notably, the complex is different from that obtained by Joachim Frank's lab (Elife 2, e00411), who observed a physical contact between RF1 and a reportedly nucleotide-free RF3.

Yes, there is always a concern when complexes are stabilized artificially, but I think the reviewer will agree that this is a common method used in structural biology, i.e. nucleotide analogs (ADPNP, GDPNP etc) as well as mutants (EQ2 mutants for ATPases etc) and in particular for ribosome structural biology, e.g. most of the translation factor-ribosome structures are stabilized with nucleotide analogs or through the use of antibiotics, such as kirromycin (EF-Tu), fusidic acid (EF-G), or a combination of (sometimes several) antibiotics and nucleotide analogs. In addition, many different tRNA states with partially rotated ribosomes have been visualized by using antibiotics, such as aminoglycosides (neomycin, viomycin, etc). In fact, fundamental insights into peptide bond formation and decoding have come from using aminoglycoside (paromomycin) and puromycin-analogs (CC-Pmn and C-Pmn), respectively. Regardless, we believe our complexes are physiologically relevant because apidaecin is a natural inhibitor of bacteria. At the minimum the complexes reflect states during the physiological inhibition by apidaecin of termination factor recycling, although we believe that they also provide insight into the natural recycling process. We find the implication that our structure is an artifact from the stabilizing methods because it is different from Joachim Franks lab to be ironic considering that their complex was formed with apo-RF3 – which has very unclear physiological relevance and more-over the map has a resolution of only 10 Å raising questions about the limits to the interpretation.

1) Is the title factually correct, given that RF3 cannot recycle RF1 in the presence of API?

We have changed the title to reflect better that apidaecin was present to “Visualization of translation termination intermediates trapped by the Api137 peptide during RF3-mediated recycling of RF1”

2) It should be made clear in the abstract exactly how this complex was trapped.

We agree and now state that “...we have used the Api137 peptide to trap RF1 together with RF3 on the ribosome and visualize an ensemble of termination intermediates using cryo-electron microscopy”.

3) The authors do not show that RF3 “mediates dissociation of RF1 indirectly by inducing SSU rotation” – the structures simply suggest that this might be possible. The language in the manuscript should be revised carefully.

The reviewer is correct. We have revised the manuscript in the direction: “We do not observe interaction between RF1 and RF3 in any of the structures, suggesting that RF3 mediates dissociation of RF1 indirectly by inducing SSU rotation.”

4) FRET experiments should be considered to show a correlation between RF1 dissociation and RF3-induced/stabilized subunit rotation. This would strengthen the proposed model.

A comprehensive FRET study on the dissociation of RF1 and RF3 and the concomitant subunit rotation is now in press in Elife and should appear online shortly. A preprint of the paper is available in the bioRxiv collection (bioRxiv 243485; doi: <https://doi.org/10.1101/243485>). The correlation between RF1 dissociation and RF3 induced subunit rotation is mentioned in the Discussion where the bioRxiv paper is already quoted (ref 22).

5) The Frank group made RF1 mutants at the interface between RF1 and RF3 that were shown to affect RF3-mediated recycling of RF1, supporting their conclusions of a physical interaction (Elife 2, e00411). These mutants should be discussed in the framework of the structures presented here – is there an alternative explanation as to why the mutations affected RF3-mediated recycling of RF1?

The mutations referred to by the reviewer are H13A and E18A, which are located in domain I of RF1. It is possible that rather than abolishing interactions at the interface of RF1 and RF3, these mutations lead to conformational changes within domain I that effect its interaction with the ribosome, as seen in state I where domain I interacts with the uL11 and uL12 of the LSU and h33 of the SSU. We note that Frank and coworkers themselves conclude that the effect of H13A is indirect: “We attribute the reduction in RF1 recycling rate by the H13A mutation to indirect—steric and/or charge—effects and propose that H13 contributes to optimal orientation of helix α 2 including E18 and other residues important for interaction with RF3. However, a direct interaction between H13 and RF3 cannot be excluded.” Therefore, using the same logic, it cannot be ruled out that the effect of the E18A

mutation (which is only 3-fold) is also indirect.” Lastly, Mora et al., 2003 (new ref 51) showed that RF3 still recycles RF1 even when RF1 is lacking domain I, suggesting any interaction/overlap between RF1 and RF3, if it does exist, is not strictly necessary for RF3-mediated RF1 recycling. This we now mention on page 9.

Reviewer 2

The molecular mechanism of translation termination in bacteria requires release factors RF1, RF2 that recognize a stop codon in the A-site and facilitate peptidyl-tRNA hydrolysis. The GTPase RF3 then releases these factors from the ribosome to allow a new translation cycle to restart. Whereas the action of RF1 and RF2 has been characterized in detail, the process of their RF3-mediated dissociation from the ribosome remained elusive, and only 9.7 Å structure of RF1 and apo-RF3 (no nucleotide form) bound to a non-rotated ribosome has been determined. In the current manuscript, Wilson and colleagues aim to describe how RF3 promotes RF1 dissociation using cryo-EM.

The authors first reconstituted a full termination complex with RF1 and then added RF3-GDPCP before subjecting it to cryo-EM. Since it resulted in a low RF1-RF3 occupancy, they cleverly supplemented the reaction mixture with API that prevents RF1 dissociation through replacing the nascent peptide in the tunnel. The consequent cryo-EM data analysis shows four distinct states resolved at 3.8-3.9 Å resolution, with RF1 and RF3 adopting different conformations. The crystal structures of tRNA, RF1 and RF3 were then fitted to the density map to interpret the data.

The series of structures show that binding of RF3-GDPCP induces rotation of the small subunit that results in RF1 dissociation. No direct interactions between RF3 and RF1 are observed. The rotation facilitates interaction between the RF3 G-domain and sarcin-ricin loop that stimulates the GTPase activity. Furthermore the rotation also leads to a shift of the P-site tRNA. Following the action of RF3, the ribosome is in the exact position to accommodate the binding of RRF and EF-G for a restart. An interesting observation is that in addition to the C-terminal domain of one copy of L7/L12, there is an extra density attributed to a second copy bridging domain I of RF1 with L11. Authors then used Δ L7/L12 strain to prove that the stalk is indeed involved in facilitating the binding of termination factors.

Overall, the study provides logical sequence of events that can serve as a working model for the recycling of release factors. A potential point of criticism is that essentially similar mechanism of RF1 release was implied before, however in my opinion the paper provides clear experimental evidence and the first direct observation of how RF3 acts on the ribosome to release RF1, which is of a high scientific value to the field of translation. The work is put well in the context, and as far as I can judge (my field is mt-translation) the previous literature is described well. Figures illustrate the reported results in a clear manner, and schematic insets are very useful for understanding, as well as the supplementary movies.

Minor comments:

- **Figure 3:** indicating hydrogen bonds at this resolution is ambitious; perhaps authors would consider adding the experimental density here as well.

It seems the reviewer overlooked Supplementary Fig. 3 panels f-i, which are devoted to images containing the electron density for the CCA-ends of the tRNAs. I agree that at this resolution we cannot observe hydrogen bonds but I think the reviewer will agree that the density is pretty clear as to how the nucleotides are oriented and since it is known that the CCA-end forms hydrogen bonds with the P-loop, it seems logical to include them. It certainly helps the understanding of the figure. In the legend, we had also written that they are potential hydrogen bonds, therefore, we have left the hydrogen bonds in the figure.

- **For the cryo-EM part,** it's a good practice to include a representing micrograph and 2D class averages.

As requested, we have now included a representative micrograph and 2D averages into Supplementary Fig. 1c.

- **Please use the updated nomenclature for ribosomal proteins (Curr opin in struct biol 24: 165-169).**

The updated nomenclature is now present in the manuscript text as well as updated in the Figures.

Reviewer 3

The manuscript by Graf et al. addresses the molecular mechanism of translation termination in prokaryotes that involves class-I release factors RF1 and RF2 and the class-II release factor RF3, which is a ribosomal GTPase. This study addresses a longstanding question of whether RF1/2 and RF3 co-exist bound to the ribosome, and how release from the ribosome is triggered mechanistically, a fundamental scientific question that several laboratories including ours tried to address in the past.

Using a novel approach with a peptide blocking RF1 dissociation (published recently by a part of the co-authors of this paper, the peptide replaces the nascent peptide in the exit tunnel), a GAQ mutant of RF1

that slows down RF1-dissociation and combining that with GDCP to block RF3, it became possible to reconstitute a more stable complex than previously without the inhibitor peptide. The structure of various sub-populations was determined by cryo-EM involving significant particle sorting and reached ~3.8 Å for the best resolved complex.

This study indeed gives interesting insights into the mechanism of translation termination. In particular, it provides the first structure of a ribosome with both RF1 and RF3 factors bound in the presence of a (non-hydrolysable) GTP analogue. Comparison of the different structures reveals ribosomal subunit rotations with distinct tRNA, RF1 and RF3 positions and conformations, revealing also tRNA translocation. The structure also suggests the presence of an unusual, shifted base-pair formation between the CCA-end of the tRNA and 23S rRNA P-loop of the peptidyl-transferase centre. Other interesting findings include the observation of the ordered switch I loop in the RF3 GTPase and the role of the L7/L12 proteins that contribute to the binding of RF1 (as shown by a complementary functional analysis using L7/L12 deletion mutants), in a manner similar to that observed previously for EF-Tu and EF-G. The fact that RF1 and RF3 co-exist on the ribosome but do not seem to interact directly suggest that the mechanism of RF1 operates indirectly through the observed conformational changes of the ribosome and the 30S head swivel movement in particular.

Overall, this study appears technically well sound and allows making interesting conclusions. It would however be good to clarify a few points.

For example, as the resolution of the maps is 3.8 Å for the best resolved complex, is the base pairing between the CCA-end of the tRNA and the P-loop resolved locally well enough to discuss it to a rather detailed extent? Is the flipping of base A73 resolved in the two complexes that are being compared? Maybe some additional picture of the structure including the cryo-EM map could be helpful, especially to ascertain the proposed register shift of the interaction between the CCA-end nucleotides and the P-loop nucleotides.

I think the reviewer overlooked the Supplementary Fig. 3 panels f-i, which are devoted to images containing the electron density for the CCA-ends of the tRNAs, including the flipped A73. We agree that the resolution is not sufficient to discuss details such as hydrogen bonds but I hope the reviewer will agree that the density is pretty clear as to how the nucleotides are oriented and since it is known that the CCA-end forms hydrogen bonds with the P-loop, it seems logical to include them in the figure (especially as it helps the understanding of the figure). In the legend, we had also written that they are potential hydrogen bonds so make no claims to see the actual hydrogen bonds.

Along the same lines, the overall structure description seems limited to domains of the factors, is the resolution not sufficient to describe more details? Possibly using cryo-EM image processing techniques like focused refinements (e.g. see series of publications in Curr Op Struct Biol 2017, von Loeffelholz et al., 2017 and references cited therein) could be helpful here and help interpret the cryo-EM maps more easily or in more detail.

As can be seen in Supplementary Fig. 2 the local resolution for the ligands is generally worse than the average resolution of the ribosome. The exceptions are perhaps the ASL of the P-tRNAs and decoding region of the RFs. The reason is clearly the flexibility of the ligands, which is generally reflective of the high conformational dynamics of the sample. The classes that we have sorted out in the manuscript enable an accurate docking of the crystal structures, which is sufficient to decipher the overall conformational changes that occur that enable us to provide a model for the mechanism of recycling. Nevertheless, we always strive for the highest resolution obtainable from each dataset, however, in this case focused refinement was not beneficial.

With these new data in hand it would now be interesting to discuss some aspects in more detail, considering for example several concepts regarding RF3 function that we described in our early work when we studied the first structure of an RF3-bound ribosome (Klaholz et al., Nature 2004) using for the first time particle sorting and revealing 2 different structures and RF3 conformations in equilibrium with each other, for example: the existence of pre- and post-translocated states (tRNA transition from the P- to the E-site plus the associated L1 movement), the RF3-dependent rotation of the ribosomal subunits, the possibility of co-existence of class-I and class-II release factors, etc. (see also our TIBS review in 2011 on translation termination, Figure 6), the fact that the G/G' domains move from an inactive RF3 conformation to an activate one towards the α -sarcin-ricin loop where GTP-hydrolysis is induced etc. Now that this can be analysed at much higher resolution it would be an opportunity to re-discuss some of these aspects.

Firstly, let us make it clear that the Klaholz et al structures were state of the art for 2004 and were a great achievement. However, the main problem we have with going into detail in relation to these structures is that the resolution was relatively low (15-25 Å). Some of the conclusions appear to be correct, i.e. different states of rotation due to RF3 – although we do not observe different conformations of RF3 in these states. Also the E-tRNA is a hybrid P/E-tRNA. RF3 does approach the SRL in the rotated state, however, the movement towards the SRL is driven by the subunit rotation, whereas in Figure 2 of Klaholz et al Nature 2004 it appears that the entire RF3 is

undergoing inter-domain conformational changes - where we see no inter-domain conformational change. For this reason, we are reluctant to “pick-and-choose” results from this paper that are in accordance with what we see. Such detailed analysis could be made in a follow-up review?

In this context it is worth mentioning that we also tried analysing the structure of a ribosome complex with RF2 and RF3 simultaneously bound, but even with structure sorting at the time it would give either RF2 or RF3 bound separately (unpublished), as also found in the present work described in the first part of the paper for RF1- and RF3-bound ribosome subpopulations. Therefore, the API inhibitor peptide is really a good and elegant approach to tackle a difficult problem and stabilize the otherwise transient complex that finally allowed addressing a longstanding question. This study will therefore be of general interest to the field.

We thank the reviewer for this comment especially given the comments of reviewer 1, who seems to be of the opinion that using any antibiotic or non-hydrolysable analog makes the complexes artificial and not physiologically relevant.

Detailed points:

- cryo-EM maps and atomic coordinates should be provided via PDB and EMD databases.

As indicated in the “Accession Numbers” section on page 20, the maps and models for States I-IV and the RF3-only structure will be deposited in the EMDB and PDB. The accession numbers will be updated in the manuscript upon acceptance.

- page 3, 3. line from bottom: typo, should say: “have so far been lacking”.

This has been corrected

- page 3, same paragraph: the concept that “Superposition of the RF3 and RF1/RF2 ribosome structures revealed no overlap of the factor binding sites” was shown in our early work already (reference 27).

The reference has been included

- page 3, last line: “binding of RF3... induces rotation of the SSU relative to the LSU” was seen in previous work also (ref. 27), this could now be discussed in the final discussion part also, as suggested above.

Yes, we already quote reference 27 for “Cryo-EM and X-ray structures exist of RF3-GDP(C/N)P (non-hydrolysable GTP analogues) bound to rotated ribosomes with P/E-hybrid state tRNAs but without the decoding release factors (Fig. 1b)^{17,27-29}.” With the assumption here that the “post-translocation state” with “E-tRNA” observed in ref 27 was actually a rotated state with a hybrid P/E-tRNA. This is not discussed in the final discussion since our paper focus is on recycling of RF1, which was not present in the structure from Ref 27.

- page 5: maybe specify how slow the RF1 mutant is compared to wild type RF1; this is particularly relevant for this study because if there was no release of the nascent peptide the inhibitor peptide would probably not be active. Apparently it is still fast enough to release the peptide because the incubation with the RF1 mutant was only 1 min as described in the methods section.

According to the literature (Ref. 7 and 9), the RF1 mutant is 3300-fold slower than the wt and as the reviewer noted, our incubation was very short. Therefore, we did not expect to observe hydrolysis and hoped to trap RF1 and RF3 on a non-rotated state. However, clearly some hydrolysis did occur, which was a benefit since we were then able to observe multiple states differing in the degree of rotation.

- page 6: linking regions of RF1 are at low resolution, maybe cryo-EM focused refinements after particle sorting could help?

We tried this, but we did not obtain significant improvements of the maps, presumably due to the continuum of states.

- page 7: does the register shift and re-base-pairing imply a bent of the tRNA acceptor stem?

There may be a small bending of the tRNA acceptor stem but the main conformational changes are a movement of the acceptor stem by 15 Å out of the PTC (see Figure 3a) and rearrangement of the flexible ACCA end (which involves the flipping of A73, which is shown we have now arrowed in Figure 3d to make it clearer.

Last phrase of this paragraph: specify that re-base-pairing also comprises a register shift, it does not re-base-pair with the same residues.

The reviewer is quite correct. The re-basepairing involves a register shift, which is the important point. We have changed this in the text on page 7 to emphasize this, as well as in the discussion on page 11.

- page 8: overall conformation of RF3 as described in references 28,29: is it also similar to one or the other sorted state of RF3 in ref. 27?

Depends what you define as similar given the limited resolution of the earlier reconstructions.

- page 8, middle: translational GTPases and activation by SRL, “such as eEF1A, SelB and EF-Tu”: true for IF2 also (e.g. Myasnikov et al., NSMB 2005; Sprink et al., Sci Adv. 2016).

True. IF2 has been included with both references.

- page 9, role of L7/L12 for the activity of ribosomal GTPases: true for IF2 also (e.g. Carlson et al., FEBS 2017; Ge et al., PNAS 2018).

True. IF2 has been included with both references.

- page 10: loss of interaction of RF1 with the P/P-tRNA: this interaction is anyway not that strong.

Agreed. For this reason, we state only that it may “contribute” to destabilization

- page 10, bottom: “...all reported structures of decoding factors RF1 and RF2 were bound to ribosomes with non-rotated conformation (ref. 14)”: this is true also for the 2 structures published by Rawat et al. and Klaholz et al., Nature 2003.

These references have been included here now.

- page 11, reference 48 suggests that RF1 leaves the ribosome after RF3, is this possible with the mechanism described here?

Unfortunately, our structures do not provide insight into the order of dissociation of the factors, as stated in the discussion on page 11.

- Fig. 1C: add label for RF1.

This has been added.

Reviewers' Comments:

Reviewer #2 (Remarks to the Author):

The authors addressed the comments, and I have no other suggestions to improve the manuscript.

Reviewer #3 (Remarks to the Author):

It is good to have provided additional EM images and use the latest nomenclature for ribosomal protein as suggested by referee 2.

Regarding local resolution of the ligands/release factors, these are indeed fine to interpret together with the known crystal structures. In the future maybe doing extensive 3D classification before focused refinements could help (focused refinements work the better the pre-sorting is).

Thanks for the discussion on previous structures of RF3 complexes. In fact the main movement of RF3 in the 2004 study (using local/focused classification) is the G and G' domain that move/rotate towards the SRL. This correlates with the ribosomal subunit rotation representing 2 pre- and post-translocation states, i.e. the SRL approach by RF3 is indeed driven by subunit rotation (or vice versa, RF3 traps one or the other ribosomal state as a function of its GTP state).

The fact that the RF1 mutant is 3300-fold slower than the wt could be included in the manuscript.

Other comments have been taken into account and the study as a whole gives nice insights into the mechanism of translation termination.

REVIEWERS' COMMENTS:

Reviewer #2 (Remarks to the Author):

The authors addressed the comments, and I have no other suggestions to improve the manuscript.

Reviewer #3 (Remarks to the Author):

It is good to have provided additional EM images and use the latest nomenclature for ribosomal protein as suggested by referee 2.

Yes, the EM images are provided in Supplementary Fig. 1c and all nomenclature has been updated as suggested by referee 2.

Regarding local resolution of the ligands/release factors, these are indeed fine to interpret together with the known crystal structures. In the future maybe doing extensive 3D classification before focused refinements could help (focused refinements work the better the pre-sorting is).

We did extensive 3D classification before focused refinement, however, the high flexibility of the sample did not allow significant improvement in the final volumes.

Thanks for the discussion on previous structures of RF3 complexes. In fact the main movement of RF3 in the 2004 study (using local/focused classification) is the G and G' domain that move/rotate towards the SRL. This correlates with the ribosomal subunit rotation representing 2 pre- and post-translocation states, i.e. the SRL approach by RF3 is indeed driven by subunit rotation (or vice versa, RF3 traps one or the other ribosomal state as a function of its GTP state).

In the 2004 study, the authors describe a change in RF3 conformation from an open to closed conformation, suggesting that this is what brings the G domain of RF3 into close proximity with the SRL, rather than the rotation of the small subunit. In our structural ensemble, we observe no conformational change in RF3 but rather the rotation of the small subunit alone, which brings the G domain of RF3 into close proximity with the SRL.

The fact that the RF1 mutant is 3300-fold slower than the wt could be included in the manuscript.

Now included on page 12

Other comments have been taken into account and the study as a whole gives nice insights into the mechanism of translation termination.

We thank the reviewer for their time and efforts that have improved the manuscript.